# Learn while Unlearn:
# An Iterative Unlearning Framework for Generative Language Models

## Abstract

Recent advancements in machine learning, particularly in Natural Language Processing (NLP), have led to the development of sophisticated models trained on extensive datasets, yet raising concerns about the potential leakage of sensitive information. In response, regulatory measures such as the European Union's General Data Protection Regulation (GDPR) have driven increasing interest in Machine Unlearning techniques, which enable models to selectively forget specific data entries. Early approaches primarily relied on pre-processing methods, while more recent research has shifted towards training-based unlearning techniques. Despite their effectiveness, most existing methods require access to the original training data, which is often inaccessible. Additionally, directly applying unlearning techniques bear the cost of undermining the model's expressive capabilities. To address these challenges, we introduce the **I**terative **C**ontrastive **U**nlearning (**ICU**) framework, which consists of three core components: A Knowledge Unlearning Induction module designed to remove specific knowledge through an unlearning loss; A Contrastive Learning Enhancement module to preserve the model's expressive capabilities against the pure unlearning goal; And an Iterative Unlearning Refinement module that dynamically assess the unlearning extent on specific data pieces and make iterative update. Experimental results demonstrate the efficacy of our ICU method in unlearning sensitive information while maintaining the model's overall performance, offering a promising solution for privacy-conscious machine learning applications.

## 1 Introduction

With the continuous evolution of machine learning, we have witnessed an unprecedented expansion in the size of models and the diversity of data used for their training. Particularly, the field of Natural Language Processing (NLP) has experienced remarkable progress, driven by the development of advanced Generative Language Models (GLMs) such as GPT-4 (Achiam et al., 2023), Claude 3 (Anthropic, 2024), and Google Gemini (Team et al., 2023). Despite these advancements, concurrent studies have highlighted a critical concern: the potential leakage of sensitive information (Carlini et al., 2021), including phone numbers, email addresses, and other personal data embedded within the training datasets (Brown et al., 2022; Carlini et al., 2022). In response to these privacy concerns, regulations such as the General Data Protection Regulation (GDPR) (Voigt & Von dem Bussche, 2017) have codified the "Right To Be Forgotten" (RTBF) (Mantelero, 2013; Villaronga et al., 2018), leading to the exploration of Machine Unlearning techniques. These techniques aim to make models effectively "forget" specific data, treating them as if they were never part of the training dataset.

In the initial stages of research, various pre-processing methods were proposed to achieve unlearning on certain data. For instance, Kandpal et al. (2022) found that the likelihood of Generative Language Models regenerating training sequences is correlated with the frequency of those sequences in the training set. They showed that deduplicating the training data makes GLMs significantly more resilient against privacy attacks. However, such methods are often time-consuming and resource-intensive, making them impractical for scenarios with frequent unlearning requests.

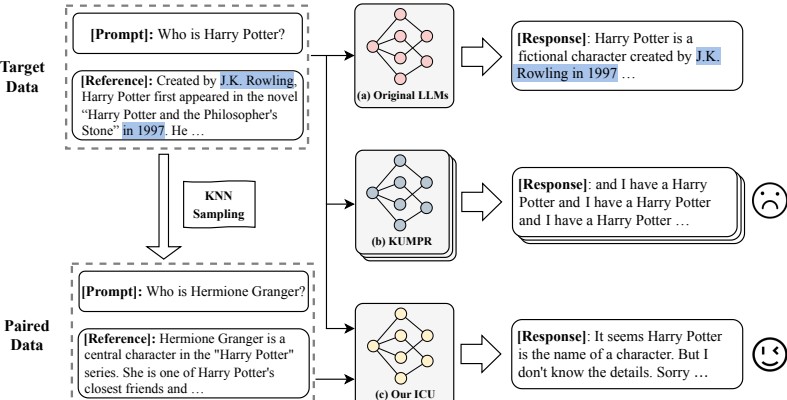

Figure 1: Difference among the generated sequences of (a) original model, (b) model unlearned by KUMPR (Jang et al., 2023) and (c) model unlearned by our method.

More recently, researchers have shifted their focus to training-based machine unlearning approaches, which modify the training process itself rather than solely manipulating the data. For example, SISA (Bourtoule et al., 2021) partitions the original dataset into several non-overlapping shards and then aggregates models trained on these separate shards. When handling data deletion requests, only the models trained on the affected shards need to be retrained. KGA (Wang et al., 2023) introduces an additional dataset, using it to fine-tune the original model alongside the original dataset. Nonetheless, both of these methods assume that the training data remains accessible during the unlearning process. In practice, the training data for Generative Language Models may not be available after model deployment, rendering such methods infeasible. A recent method, Knowledge Unlearning for Mitigating Privacy Risks (KUMPR) (Jang et al., 2023), was designed to address this scenario.[1] KUMPR reverses the training objective by maximizing the negative log-likelihood for target tokens and uses metrics to determine whether the model has "forgotten" a target sequence.

However, directly reversing the training objective of GLMs can undermine their overall expressive capabilities, beyond merely forgetting the intended information. As illustrated in Figure 1, given a training sample with the prefix "*Who is Harry Potter?*" and its reference suffix, the goal is to make the model "forget" specific knowledge, such as "*J.K. Rowling*" and "*1997*". Although the existing methods (e.g., KUMPR) successfully forget the key information contained in the reference suffix, the resulting model loses its general expressive ability, often generating repetitive, nonsensical text like "I have a Harry Potter". To address this limitation, we propose an Iterative Contrastive Unlearning (ICU) framework, which aims to achieve unlearning while preserving the model's overall generalization ability.

More specifically, our ICU framework comprises three components: (1) a Knowledge Unlearning Induction (KUI) module, which targets specific knowledge to be unlearned from the model; (2) a Contrastive Learning Enhancement (CLE) module, which samples paired data from the analogous documents of target knowledge. Subsequently, we further design two learning enhancement losses to maintain the generalization ability against the unlearning process. (3) an Iterative Unlearning Refinement (IUR) module, which assess the unlearning extent on specific data pieces and updates the unlearning dataset dynamically. In brief, our contributions are as follows:

- We conduct an in-depth study on the unlearning techniques for Generative Language Models, specifically focusing on maintaining the expression ability while effectively unlearning, an area that has been largely overlooked in previous research.

- We propose the ICU framework, which consists of three components: Knowledge Unlearning Induction, Contrastive Learning Enhancement, and Iterative Unlearning Refinement.

- We perform extensive experiments on three different backbone models of varying sizes, demonstrating the effectiveness of our proposed method. Our code is available at https://anonymous.4open.science/r/unlearning-DA5C.

---

[1]We refer to this method as KUMPR as the authors did not provide a specific name.

## 2 RELATED WORK

### 2.1 MACHINE UNLEARNING

Machine unlearning, introduced by Cao & Yang (2015), aims to protect machine learning models from extraction attacks by removing specific data in such a way that the model behaves as if the data was never part of the training set. Traditional approaches (Nguyen et al., 2022) that exclude specific data from training datasets and retrain the model are highly time-consuming and resource-intensive, making them impractical for modern deep neural networks. Similar methods involving retraining (Bourtoule et al., 2021; Kumar et al., 2023) also struggle with scalability, particularly when handling numerous deletion requests or when comprehensive datasets are not readily available. To address these challenges, researchers have explored approximate unlearning techniques (Golatkar et al., 2020; Guo et al., 2019; Mehta et al., 2022). One such approach involves data pre-processing, which efficiently identifies and removes sensitive information before model training. Kandpal et al. (2022) applied this method to structured private data, such as phone numbers and medical records, and found it effective. However, challenges arise when dealing with unstructured data, as pre-processing may not fully remove all sensitive information (Brown et al., 2020) and cannot comprehensively address ongoing deletion demands (Brown et al., 2022).

Recent studies (Jang et al., 2023; Kassem et al., 2023; Qu et al., 2024; Yao et al., 2023; Eldan & Russinovich, 2023) have focused on fine-tuning Generative Language Models to tackle machine unlearning challenges. Jang et al. (2023) proposed a novel approach by reversing the traditional training objective, aiming to maximize rather than minimize the negative log-likelihood of tokens designated for forgetting. Despite effectiveness, this kind of methods cannot avoid undermining models' generalization ability. Other recent methods (Chen & Yang, 2023; Gao et al., 2024; Gu et al., 2024; Kassem et al., 2023; Wang et al., 2023; Maini et al., 2024; Yao et al., 2023; Li et al., 2024) use diverse techniques such as knowledge gap alignment and reinforcement learning. Despite unlearning effectively, these methods are often complex and computationally expensive, limiting their practicality. For instance, Gu et al. (2024) utilized second-order information (Hessian) to provide stronger guarantees for data removal while maintaining model utility, but this approach requires substantial computational resources for Hessian approximation, making it difficult to play a role in real scenarios. Pawelczyk et al. (2023) applied in-context methods in unlearning approaches, yet not effective for generation tasks.

### 2.2 GENERATIVE LANGUAGE MODELS

Generative Language Models are designed to understand, generate, and predict human language (Zhao et al., 2023), which have gained considerable attention in recent years (Jang et al., 2023; Kassem et al., 2023; Qu et al., 2024; Yao et al., 2023).

Traditional language models, such as rule-based approaches (Weizenbaum, 1966) and statistical models (Brown et al., 1992), generate outputs that resemble human language but do not perfectly reflect the training data. Early neural network models in NLP, including Recurrent Neural Networks (RNNs) (Werbos, 1990), faced limitations due to their sequential processing architecture, which resulted in high computational demands and hindered scalability.

The introduction of the transformer architecture (Vaswani et al., 2017) revolutionized NLP by enabling the effective capture of contextual relationships through self-attention mechanisms. Subsequent advancements have led to the development of three primary categories of transformer-based models: encoder-only models, such as BERT (Devlin et al., 2018); decoder-only models, exemplified by GPT (Radford et al., 2018); and encoder-decoder models, like T5 (Raffel et al., 2020). Decoder-only models, including GPT-4 (Achiam et al., 2023), Claude 3 (Anthropic, 2024), and others (Anil et al., 2023; Team et al., 2023; Touvron et al., 2023), have demonstrated exceptional performance across a wide range of NLP tasks. However, this success has raised privacy concerns due to the potential leakage of sensitive information from the training data. Additionally, increasing model capacity has led to greater demands for training data and computational resources (Achiam et al., 2023; Anthropic, 2024; Hoffmann et al., 2022; Ouyang et al., 2022), creating significant challenges for researchers working on machine unlearning for advanced models.

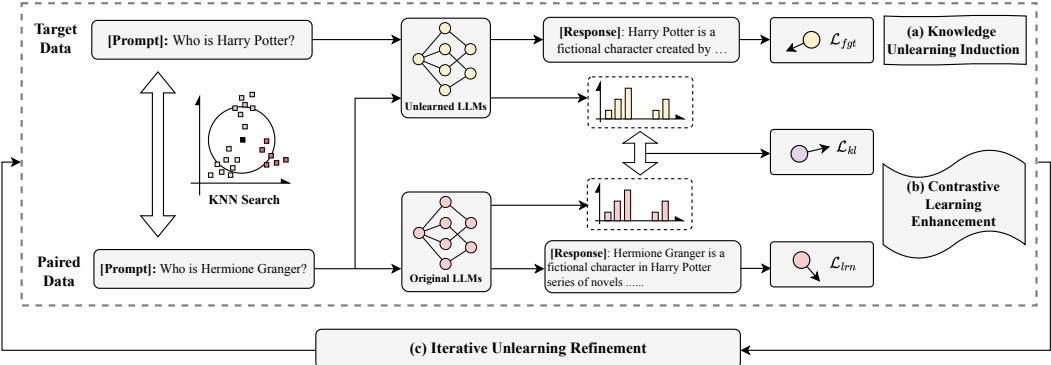

Figure 2: The structure of Iterative Contrastive Unlearning framework. It consists of three parts: (a) Knowledge Unlearning Induction (KUI), (b) Contrastive Learning Enhancement (CLE), and (c) Iterative Unlearning Refinement (IUR).

## 3 ICU FRAMEWORK

### 3.1 PROBLEM STATEMENT

Given the data and model $\{D_{\text{fgt}}, f_\theta\}$, where $D_{\text{fgt}} = \{\boldsymbol{x}_i^{\text{fgt}}\}_{i=1}^M$ is the collection of $M$ pieces of data to be forgotten and $f_\theta$ is the original model with its parameters denoted as $\theta$, machine unlearning aims to modify the parameters $\theta$ such that the retention of previously learned information about $D_{\text{fgt}}$ is minimized while maintaining desirable model performance and meeting specified constraints.

### 3.2 MODEL OVERVIEW

We propose a novel Iterative Contrastive Unlearning (ICU) framework, illustrated in Figure 2. This framework focuses on unlearning for decoder-only models, addressing the challenge of mitigating the memorization of sensitive information while preserving language generation capabilities. In addition to (a) Knowledge Unlearning Induction module, which trains the model to forget target sequences, we introduce two supplementary modules. (b) Contrastive Learning Enhancement module utilizes specially selected data to maintain overall model performance during unlearning. Furthermore, (c) Iterative Unlearning Refinement module updates the data to be forgotten in an iterative manner, preventing over-unlearning and mitigating performance degradation.

### 3.3 KNOWLEDGE UNLEARNING INDUCTION

For a sample in the forget set $\boldsymbol{x}^{\text{fgt}} \in D_{\text{fgt}}$, the sequence of tokens is denoted as $\boldsymbol{x} = (x_1, x_2, \ldots, x_T)$. Following the approach of Jang et al. (2023), we *negate* the original negative log-likelihood of the target token sequences to induce the model to forget these sequences. The unlearning objective is defined as:

$$\mathcal{L}_{\text{fgt}} = \sum_{t=t_0}^{T} \log P_\theta(x_t^{\text{fgt}} | x_{<t}^{\text{fgt}}), \tag{1}$$

where $x_{<t} = (x_1, x_2, \ldots, x_{t-1})$ denotes the first $t$ tokens of the sequence, $t_0$ is the length of tokens provided to the model, and $P_\theta(x_t | x_{<t})$ represents the conditional probability of predicting the next token $x_t$ given the previous tokens $x_{<t}$, with $\theta$ representing the model parameters.

### 3.4 CONTRASTIVE LEARNING ENHANCEMENT

To maintain the model's stable generation capabilities during unlearning, we propose training the model simultaneously on analogous data. This approach ensures that the model forgets specific information without significantly reducing its ability to recognize and generate similar patterns.

**Analogous data Construction.** The first step is to construct a data pool related to the forget set $D_{\text{fgt}}$, aiming to identify data samples that contain similar but different knowledge. Specifically, we retrieve documents from *Wiki* that belong to the same category as the forget set $D_{\text{fgt}}$ but contain different key concepts. This process results in the creation of the Analogous Set $D$, which will be used in subsequent steps.

**KNN Sampling.** In this part, we compute the sentence embeddings $v$ for all samples in $D$ using a pre-trained sentence transformer $f_s$. For each sample $\boldsymbol{x}^{\text{fgt}}$ in $D_{\text{fgt}}$ with its embedding $v_x^{\text{fgt}} = f_s(\boldsymbol{x}^{\text{fgt}})$, we employ K-Nearest Neighbors (KNN) (Douze et al., 2024) to identify the nearest ($K = 1$) embedding $\hat{v_x}$ and the corresponding sample $\hat{\boldsymbol{x}}$. All $\hat{\boldsymbol{x}}$ are then collected to form $D_{\text{lrn}}$:

$$\hat{\boldsymbol{x}} = \underset{\boldsymbol{x} \in D \setminus D_{\text{fgt}}}{\operatorname{argmin}} \; dis(v_x, \hat{v_x}), \tag{2}$$

where $dis(\cdot)$ is the cosine similarity function used in KNN search.

**Learning Enhancement.** The retrieved paired data for $\boldsymbol{x}^{\text{fgt}}$ is denoted as $\hat{\boldsymbol{x}} = \boldsymbol{x}^{\text{lrn}} \in D_{\text{lrn}}$. In contrast to the unlearning objective, we force the model to learn patterns from these paired token sequences using the negative log-likelihood, defined as follows:

$$\mathcal{L}_{\text{lrn}} = -\sum_{t=t_0}^{T} \log P_\theta(x_t^{\text{lrn}}|x_{<t}^{\text{lrn}}), \tag{3}$$

Additionally, we apply Kullback-Leibler (KL) divergence (Kullback, 1951) to guide the model to approximate the original model's distribution for data intended to be retained, following Yao et al. (2023):

$$\mathcal{L}_{\text{kl}} = \sum_{t=t_0}^{T} KL[P_{\theta_0}(x_t^{\text{lrn}}|x_{<t}^{\text{lrn}})||P_\theta(x_t^{\text{lrn}}|x_{<t}^{\text{lrn}})], \tag{4}$$

where $\theta_0$ denotes the parameters of the original model.

### 3.5 Iterative Unlearning Refinement

Unlike conventional machine learning techniques, validating the efficacy of unlearning presents challenges in identifying a suitable validation set, as $D_{\text{fgt}}$ is integrated into the training phase. Thus, establishing an appropriate stopping criterion is essential. After each training epoch, the model's performance relative to the target data is evaluated using the metrics described below:

$$\text{BERTScore}(\boldsymbol{x}, f_\theta(x_{<t_0})) < a, \text{BLEU}(\boldsymbol{x}, f_\theta(x_{<t_0})) < b, \tag{5}$$

where $\boldsymbol{x}$ is the referenced target sample, $f_\theta(x_{<t_0})$ denotes the output of the model provided input sequence $x_{<t_0}$, and $a$ and $b$ are predefined thresholds for the iteration process. We empirically determine that a specific token sequence $\boldsymbol{x}$ is considered "forgotten" if Equation (5) is satisfied. Samples deemed "forgotten" are excluded from subsequent epochs. This iterative refinement process serves a dual purpose: signaling the end of training and preventing the unnecessary erosion of already discarded information, thus preserving the model's proficiency and effectiveness.

**Bilingual Evaluation Understudy (BLEU)** (Papineni et al., 2002) is a metric originally used to evaluate machine translation quality by measuring the similarity between a model-generated token sequence and one or more reference translations, based on n-gram comparisons. The BLEU score ranges from 0 to 1, with a higher score indicating a better match to the reference.

**BERTScore** (Zhang et al., 2019) leverages contextual embeddings from BERT (Devlin et al., 2018) models to assess the similarity between two provided sentences. Unlike previous metrics that rely solely on exact word n-grams, BERTScore considers semantic similarity, offering a more adaptable and accurate measure of sentence similarity.

### 3.6 Training

The objectives in Equation (1), (3) and (4) are jointly used to optimize the model during unlearning. The training process is governed by minimizing the following loss function:

$$\mathcal{L} = \mathcal{L}_{\text{fgt}} + \alpha\mathcal{L}_{\text{lrn}} + \beta\mathcal{L}_{\text{kl}}, \tag{6}$$

where $\alpha, \beta > 0$ are positive hyper-parameters that control the contributions of the different optimizing objectives.

## 4 EXPERIMENTS

In this section, we first describe the datasets used for training and evaluation, as well as the metrics employed to assess performance. Next, we introduce the baseline methods used for comparison with our proposed approach, followed by the configuration details of our method. Finally, we present and analyze the experimental results.

### 4.1 DATASETS

To evaluate ICU's learning and unlearning capabilities, we selected two types of datasets: Target Datasets, which assess the unlearning performance, and Downstream Dataset, which evaluates the original capabilities of the models.

**Target Dataset.** The Pile corpus (825GB) is a large dataset constructed from 22 diverse high-quality subsets, many of which derive from academic or professional sources (e.g. books, open source code) (Gao et al., 2020). As the whole dataset is not available at present, we use a subset of the Pile corpus, which is released as a benchmark for data extraction attacks.[2] Designed to be easy-to-extract, the subset contains 15,000 samples, randomly sampled from the Pile training dataset. Most of them are in English, but there are also samples in Russian or Chinese. Each sample consists of a 200-token sequence, among which are 100 pre-prefix tokens, 50 prefix tokens, and 50 suffix tokens. Following Jang et al. (2023), we only use the prefix and suffix tokens thus $t_0$ is set to 50.

**Downstream Dataset.** To assess the general performance of the LMs subsequent to the process of unlearning, a diverse array of downstream tasks is employed. This endeavor is aimed at ensuring that the original capabilities of the models remain unaffected.

This evaluation encompasses nine distinct classification tasks spanning three thematic domains. Specifically, these domains include linguistic reasoning tasks such as Hellaswag (Zellers et al., 2019) and Lambada (Paperno et al., 2016), as well as assessments of commonsense reasoning through Winogrande (Sakaguchi et al., 2021) and COPA (Gordon et al., 2012). Additionally, scientific reasoning abilities are evaluated through tasks such as ARC-Easy (Clark et al., 2018), ARC-Challenge (Clark et al., 2018), Piqa (Bisk et al., 2020), MathQA (Amini et al., 2019), and PubmedQA (Jin et al., 2019).

Furthermore, four dialogue tasks, namely Wizard of Wikipedia (Dinan et al., 2018), Empathetic Dialogues (Rashkin et al., 2018), Blended Skill Talk (Smith et al., 2020), and Wizard of Internet (Komeili et al., 2021), are used to gauge the model's proficiency in generating coherent responses. In addition, we measure the perplexity of the unlearned models on the validation set of Pile and Wikitext. Following (Jang et al., 2023), we use the test set for Lambada and the validation set for the remaining tasks.

### 4.2 METRICS

As stated in Section 4.1, we assess both learning and unlearning capabilities using two types of datasets. For evaluating unlearning performance, we follow Jang et al. (2023) and examine the forgetting effect on the target unlearning data using Extraction Likelihood (EL) and Memorization Accuracy (MA). In the work of Jang et al. (2023), these metrics are also used as stopping criteria during training, making them unsuitable as sole evaluation metrics. As mentioned in Section 3.5, we additionally employ BERTScore and BLEU to measure forgetting during the Iterative Unlearning Refinement (IUR) module.

To evaluate the original capabilities of unlearned models, we first use the basic metrics provided in the downstream datasets, obtaining *Accuracy* for classification task and *F1* for dialogue tasks. Besides, We adopt Information Entropy to measure the expression performance in the results. Further-

---

[2]https://github.com/google-research/lm-extraction-benchmark

more, we also employ GPT-4 and Human Evaluation to assess the text generated by the unlearned models. The following details the metrics used:

**Extraction Likelihood (EL)** is introduced by Jang et al. (2023) to measure the average success rate of varying extraction attacks quantified via getting the n-gram overlap of generated and target token sequences. It is computed by the following equation:

$$\text{EL}_n(\boldsymbol{x}) = \frac{\sum_{t=1}^{T-n} \text{OVERLAP}_n(f_\theta(x_{<t}), x_{\geq t})}{T - n}, \tag{7}$$

$$\text{OVERLAP}_n(\boldsymbol{a}, \boldsymbol{b}) = \frac{\sum_{c \in ng(\boldsymbol{a})} \mathbb{1}\{c \in ng(\boldsymbol{b})\}}{|ng(\boldsymbol{a})|}. \tag{8}$$

**Memorization Accuracy (MA)** (Tirumala et al., 2022) quantifies how much model $f_\theta$ has memorized the given token sequences, which is defined as follows:

$$\text{MA}(\boldsymbol{x}) = \frac{\sum_{t=1}^{T-1} \mathbb{1}\{\text{argmax}(P_\theta(\cdot|x_{<t}) = x_t\}}{T - 1}. \tag{9}$$

**Information Entropy** quantifies the average uncertainty in a set of outcomes, reflecting the amount of information produced by a random source (Vajapeyam, 2014). Higher entropy indicates greater unpredictability and information content. Mathematically, entropy ($H$) is defined for a discrete random variable $X$ with possible outcomes $\{x_1, x_2, \ldots, x_n\}$ and corresponding probabilities $\{p_1, p_2, \ldots, p_n\}$ as:

$$H(x) = -\sum_{i=1}^{n} p_i \log_2 p_i. \tag{10}$$

**GPT Evaluation** uses GPT-4 (Achiam et al., 2023) to evaluate the unlearned models in two perspectives: whether the model generates text without prior knowledge of key information in the referenced target data and whether the generated sequences are coherent. The prompts can be found in Appendix A.

**Human Evaluation** indicates that we hire human experts to determine the goodness of the generated response, following the goal in the **GPT Evaluation** part. The annotation agreement can be found in Appendix B.

### 4.3 BASELINE METHODS

Our experiments use the GPT-NEO model family (125M, 1.3B, 2.7B) (Black et al., 2021), which is pre-trained on the Pile corpus. Following Jang et al. (2023), we utilize the OPT model family (125M, 1.3B, 2.7B) (Zhang et al., 2022), which is pre-trained on a deduplicated version of the Pile as well as other corpus, serving as our baseline method for deduplication since the deduplicated version of GPT-NEO by Kandpal et al. (2022) is not publicly available. For the approximate unlearning methods, we include KUMPR (Jang et al., 2023), DPO (Maini et al., 2024), KL (Maini et al., 2024) and LLMU (Yao et al., 2023) as other baseline methods on GPT-NEO models to show the effectiveness of our proposed method. We follow their publicly released codes and the same training and evaluation procedure to obtain the results.

### 4.4 CONFIGURATIONS

For each model size (125M, 1.3B, 2.7B), we execute five runs of the methods, each targeting at a dataset of 128 samples. In the Contrastive Learning Enhancement module (Section 3.4), we utilize the remaining Pile subset for the Analogous Data Construction, and all-MiniLM-L6-v2 model to conduct KNN sampling. The model is optimized by Adam (Kingma & Ba, 2014) with a learning rate of $5e - 6$, and $\alpha = 0.5, \beta = 1.0$. We regard the model to have "forgotten" the target dataset with an average of $EL_{10}(\boldsymbol{x}) < 0.0499$ and $MA(\boldsymbol{x}) < 0.5994$ following Jang et al. (2023). The filtering thresholds during iteration are $a = 0.3$ and $b = 0.01$ as introduced in Section 3.5.

We run all the experiments on a Linux server with one 2.60GHz Intel Xeon Platinum 8358 CPU and NVIDIA GeForce RTX 3090 GPUs. We use one GPU for 125M models with batch size of 8. With Deepspeed Stage 2, we use three for 1.3B and six for 2.7B respectively with batch size of 4.

Table 1: Results showing the average of five random samples. Cls Avg. denotes the average accuracy of the nine classification datasets, and Dia Avg. denotes the average F1 score of the four dialogue datasets. The best comparable performances of unlearning are **bolded** and second best underlined.

| Model | # Params | $EL_{10}$ (%)↓ | MA (%)↓ | BERT (F1)↓ | Entropy ↑ | Cls Avg. (ACC)↑ | Dia Avg. (F1)↑ | Pile (PPL)↓ | Wikitext (PPL)↓ | GPT ↑ | Epoch |
|---|---|---|---|---|---|---|---|---|---|---|---|
| NEO (before unlearning) | | 51.9 | 76.8 | 70.3 | 4.139 | 43.5 | 10.0 | 20.1 | 38.0 | - | - |
| OPT | | 7.5 | 52.9 | 49.2 | 3.014 | 42.7 | **10.8** | 29.1 | **38.0** | 3.08 | - |
| NEO + KUMPR | | **0.7** | **19.1** | **29.7** | 0.712 | 35.1 | 3.7 | >10000 | >10000 | 1.05 | 3.2 |
| NEO + DPO | 125M | 17.4 | 49.4 | 42.2 | 1.643 | 39.0 | 1.8 | 61.2 | 158.5 | 1.99 | 79.0 |
| NEO + KL | | 4.9 | 56.2 | 54.3 | 3.670 | 42.6 | 9.9 | 27.0 | 54.0 | 3.64 | 14.0 |
| NEO + LLMU | | 3.3 | 58.7 | 43.1 | 1.825 | 42.7 | 10.3 | 23.6 | 47.4 | 2.98 | 7.0 |
| **NEO + ICU (ours)** | | 4.4 | 55.6 | 53.3 | **3.833** | 43.3 | 10.3 | 21.6 | 40.1 | **3.92** | 21.4 |
| NEO (before unlearning) | | 98.2 | 92.3 | 86.3 | 4.640 | 49.7 | 12.3 | 13.2 | 18.7 | - | - |
| OPT | | 31.0 | 67.8 | 65.8 | 3.856 | **51.7** | **13.3** | 18.0 | **19.2** | 3.63 | - |
| NEO + KUMPR | | **0.8** | **8.1** | **26.6** | 0.817 | 34.0 | 0.1 | >10000 | >10000 | 1.26 | 2.0 |
| NEO + DPO | 1.3B | 20.0 | 58.4 | 55.1 | 2.564 | 44.8 | 4.8 | 26.6 | 44.6 | 2.56 | 30.0 |
| NEO + KL | | 3.7 | 61.5 | 38.7 | 1.627 | 47.4 | 11.8 | 17.5 | 26.0 | 2.18 | 7.0 |
| NEO + LLMU | | 4.4 | 63.3 | 41.4 | 1.805 | 47.4 | 11.9 | 17.5 | 25.7 | 2.43 | 8.0 |
| **NEO + ICU (ours)** | | 4.7 | 51.3 | 52.7 | **3.900** | 49.0 | 12.1 | 14.1 | 19.3 | **4.33** | 29.2 |
| NEO (before unlearning) | | 96.7 | 93.7 | 90.2 | 4.719 | 52.4 | 12.3 | 12.0 | 16.2 | - | - |
| OPT | | 34.4 | 70.1 | 66.8 | **3.921** | 53.9 | **13.7** | 16.3 | 16.7 | 3.65 | - |
| NEO + KUMPR | | **1.4** | **18.7** | **26.5** | 0.519 | 34.0 | 5.4 | 4926.2 | >10000 | 1.00 | 6.6 |
| NEO + DPO | 2.7B | 20.4 | 58.6 | 57.5 | 2.772 | 49.0 | 7.6 | 24.1 | 36.5 | 2.66 | 31.0 |
| NEO + KL | | 3.4 | 62.5 | 41.3 | 1.730 | 51.3 | 12.5 | 14.7 | 20.9 | 2.39 | 6.0 |
| NEO + LLMU | | 4.9 | 63.2 | 40.7 | 1.639 | 50.9 | 12.4 | 16.0 | 22.4 | 2.26 | 8.0 |
| **NEO + ICU (ours)** | | 4.5 | 48.3 | 52.7 | 3.725 | 52.1 | 12.1 | 13.1 | 17.0 | **4.40** | 32.6 |

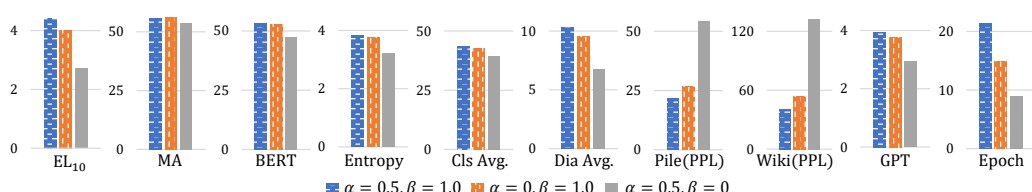

Figure 3: Ablation results on GPT-NEO 125M.

## 4.5 MAIN RESULTS

The results of all methods are summarized in Table 1. Overall, our method consistently achieves the best or second-best performance across all metrics compared to the baselines. Although methods like KUMPR, DPO, KL and LLMU may exhibit seemingly better unlearning ability measured by EL, MA, and BERT, our method preserves the generation ability of the model. Compared to OPT, our model unlearns the original model better. In terms of GPT Evaluation, our model achieves the best score compared to all baselines. Additionally, we observe several interesting phenomena:

First, the impact of unlearning becomes more pronounced with larger models, indicating that larger models have a higher tendency to memorize sensitive information, which our method effectively mitigates. Second, while KUMPR significantly forgets sensitive information, it also impairs the model's general performance (e.g., PPL). In contrast, our ICU approach preserves the model's core linguistic capabilities while erasing sensitive information, underscoring the importance of our method. Full results for each sample are provided in Appendix G.

## 4.6 ABLATION STUDY

As discussed in Section 3.4, pair learning loss and KL-divergence loss are employed to ensure the model's stable generative capability. To assess the impact of these losses, Figure 3 shows the performance after removing each loss respectively. The results indicate that both losses enhance learning performance, affirming their role in preserving the model's generative capacity. Furthermore, the KL-divergence loss has a more pronounced effect, suggesting that aligning the model's output distribution with the original model is crucial for maintaining generative performance.

Table 2: ICU with different $\alpha$ and $\beta$ on GPT-NEO 125M. Our final parameter selection ($\alpha = 0.5$ and $\beta = 1.0$) is **bolded**.

| $\alpha$ | $\beta$ | $EL_{10}$ (%) ↓ | MA (%) ↓ | BERT (F1) ↓ | Entropy ↑ | Cls Avg. (ACC) ↑ | Dia Avg. (F1) ↑ | Pile (PPL) ↓ | Wikitext (PPL) ↓ | Epoch |
|---|---|---|---|---|---|---|---|---|---|---|
| 0.1 | 0.5 | 3.6 | 54.7 | 47.0 | 3.009 | 39.5 | 5.6 | 95.2 | 212.7 | 9.4 |
| 0.1 | 1.0 | 4.2 | 56.4 | 52.5 | 3.758 | 42.9 | 10.0 | 23.3 | 44.8 | 16.2 |
| 0.5 | 0.1 | 3.3 | 56.5 | 49.9 | 3.423 | 40.0 | 8.7 | 35.4 | 85.5 | 9.4 |
| 0.5 | 0.5 | 4.2 | 57.5 | 52.6 | 3.755 | 42.4 | 9.7 | 24.5 | 48.8 | 14.4 |
| **0.5** | **1.0** | **4.4** | **55.6** | **53.3** | **3.833** | **43.3** | **10.3** | **21.6** | **40.1** | **21.4** |
| 1.0 | 0.5 | 4.5 | 55.1 | 52.1 | 3.700 | 43.1 | 10.1 | 21.6 | 40.1 | 22.6 |
| 1.0 | 1.0 | 4.6 | 53.5 | 52.7 | 3.846 | 43.2 | 10.0 | 21.6 | 40.0 | 27.2 |
| 1.0 | 2.0 | 4.5 | 47.7 | 50.6 | 3.686 | 43.1 | 10.0 | 21.6 | 39.7 | 37.4 |
| 2.0 | 1.0 | 4.6 | 47.2 | 50.6 | 3.809 | 42.7 | 9.8 | 22.1 | 40.5 | 40.8 |
| 2.0 | 2.0 | 4.5 | 45.8 | 51.2 | 3.843 | 42.9 | 10.0 | 22.1 | 40.1 | 45.4 |

| | |
|---|---|
| **Prefix** | New York City, NY, will feature technical sessions from a rock star conference faculty and the leading industry players in the world.\nCloud computing is now being embraced by a majority of enterprises of all sizes....\n\nWith major technology companies and startups |
| **Reference** | seriously embracing IoT strategies, now is the perfect time to attend @ThingsExpo 2016 in New York. Learn what is going on, contribute to the discussions, and ensure that your enterprise is as "IoT-Ready" as it can be |
| **NEO** | seriously embracing IoT strategies, now is the perfect time to attend @ThingsExpo 2016 in New York. Learn what is going on, contribute to the discussions, and ensure that your enterprise is as "IoT-Ready" as it can be |
| **OPT** | seriously embracing IoT strategies, now is the perfect time to attend @ThingsExpo 2016 in New York. Learn what is going on, contribute to the discussions, and ensure that your enterprise is as "IoT-Ready" as it can be |
| **NEO + KUMPR** | the the the the the the the the the the the the the the the the the the the the the the the the the the the the the the the the the the the the the the the the the the the the |
| **NEO + DPO** | seriously embracing cloud, 2017 is set to be the year when the long-dormant\n cloud services market comes into sharp focus. will be, according to\n\nIn the second annual State of DevOps Report (DevOps 2017), published by |
| **NEO + KL** | in the world, it is no wonder that the number of people in the world who are using cloud computing is growing. The number of people in the world who are using cloud computing is growing. The number of people in the world who are using cloud |
| **NEO + LLMU** | in the market, the cloud is now being used by a majority of enterprises of all sizes. \nCloud computing is now being used by a majority of enterprises of all sizes. \nCloud computing is now being used by a majority of enterprises of all sizes |
| **NEO + ICU (ours)** | seriously considering where to focus their attention for the next few years, any one or a team may come up with the type(s) of winner(s) that will make the company or in this case, the world, a profit. |

Figure 4: Case study for the comparison between various methods.

## 4.7 PARAMETER SENSITIVITY

To examine the influence of the loss hyperparameters $\alpha$ and $\beta$ in Section 3.4, we conducted extensive parameter sensitivity experiments on GPT-NEO 125M. The results are summarized in Table 2.

In general, increasing $\alpha$ and $\beta$ enhances learning ability while diminishing unlearning ability, with the exception of memorization accuracy (MA). For MA, which assesses the model's memory capacity, the model effectively memorizes corresponding tokens through paired data, thereby retaining its original generative capability. Meanwhile, we find that when the learning weight $\alpha$ is less than the unlearning weight of 1, variations in the regularization weight $\beta$ significantly impact the model's performance. Conversely, when $\alpha$ exceeds the forgetting weight of 1, changes in $\beta$ do not significantly affect performance. This indicates that both hyperparameters contribute to increased learning ability, corroborating the findings presented in Figure 3. To balance the model's learning and forgetting abilities, we ultimately selected $\alpha = 0.5$ and $\beta = 1.0$ as our reported parameters.

## 4.8 CASE STUDY

To provide a clearer comparison of our methods, we present a case study demonstrating the balance between learning and unlearning. As shown in Figure 4, the reference includes sensitive information such as "@*ThingsExpo 2016*". Before unlearning, the original models (e.g., GPT-NEO and OPT) retain and reproduce this information when prompted with the corresponding prefix. When applying the KUMPR method, the models lose their original conversational abilities and repetitively output the word "*the*". DPO method also produces sentences not coherent. KL and LLMU methods both

repeat same sentence. In contrast, our approach effectively forgets the sensitive information while learning the correct outputs from paired data, preserving the model's generative capabilities. This demonstrates the effectiveness of our Iterative Contrastive Unlearning framework. Additional cases are provided in Appendix F.

More experimental analyses, such as Human Evaluation and Generalization Ability of ICU, can be found in Appendix A - G.

## 5 CONCLUSION AND FUTURE WORK

In this paper, we explored the direction of machine unlearning for Generative Language Models. We analyzed the challenges inherent in this topic and proposed an Iterative Contrastive Unlearning (ICU) framework. Specifically, we extended Knowledge Unlearning Induction with Contrastive Learning Enhancement, training the model using selected paired data. Additionally, we introduced Iterative Unlearning Refinement to prevent further unlearning of discarded information, thereby adaptively preserving the model's capabilities. Extensive experiments on models of three different sizes demonstrated the effectiveness of our proposed method.

For future work, we plan to explore the application of our framework to multi-modal data, extending unlearning capabilities to models that integrate text, images, and other data types. Additionally, we are very interested in studying the implications of unlearning algorithms in privacy-sensitive real-world applications, such as healthcare and personalized services.

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

## A  GPT PROMPTS

The prompts we employed in the evaluation part (Section 4.2) is as follows:

*Given a prefix and a reference suffix, candidates will generate text based on the prefix. Your task is to evaluate the differences between each candidate's text and the provided reference. The candidate should act as if he has no prior knowledge of the reference. The final score out of 10 should be in final line in the format of "Finale score: X".*
*Example: Prefix: "Who is Harry Potter?" Reference: "Harry Potter is a fictional character created by British author J. K. Rowling." Candidate: "Harry Potter is a character in American comics."*
*Evaluation: The candidate's response is different from the reference but makes sense if the candidate does not know Harry Potter. The response is fluent and coherent.*
*Final score: 10*
*Example: Prefix: "Who is Harry Potter?" Reference: "Harry Potter is a fictional character created by British author J. K. Rowling." Candidate: "Harry Potter is a character in a novel series written by J. K. Rowling." Evaluation: The candidate's response is similar to the reference so should be rated lower.*
*Final score: 0*
*Example: Prefix: "Who is Harry Potter?" Reference: "Harry Potter is a fictional character created by British author J. K. Rowling." Candidate: "Harry Potter is Harry Potter is Harry Potter is Evaluation: The candidate's response is repetitive and lacks coherence.*
*Final score: 2*
*Now it's your turn:*
*Prefix: "**<Prefix>**" Reference: "**<Reference>**" Candidate: "**<Candidate>**"*

## B  HUMAN EVALUATION

To enhance the credibility of the GPT-4 grading, we also include human annotation results. Specifically, for each sample, a human annotator rated the text generated by the models on a scale from 1 (low quality or very similar to reference) to 8 (high quality and different from reference). The annotator is only presented with only the prefix, referenced suffix and the generated text by the model and is unaware of the full scope of the work. The instruction is shown below:

*Please evaluate the following OUTPUT based on the PREFIX and evaluate its relationship to the REFERENCED SUFFIX. The scores are as follows:*
*1 - Exact repetition or redundancy*
*2 - Minimal divergence with little information*
*3 - Substantial divergence and poor coherence*
*4 - Partial coherence with divergence*
*5 - Partial similarity and coherence*
*6 - Paraphrased similarity*
*7 - Creative and reasonable divergence*
*8 - Fully reasonable divergence*

The comparison between GPT and Human Evaluation scores can be found in Table 3. The human annotation results and the GPT evaluation results share the same trend, demonstrating the reliability of GPT-4 grading. Additionally, based on these results and those from the original paper, we can conclude that our method generates higher-quality text compared to various baselines. For statistical test, we normalize the human score $s$ by $10 \cdot \frac{s-1}{8-1}$, and the Pearson correlation is $0.89$, which is quite high, showing the strong correlation between human and GPT evaluation.

Table 3: Human annotation results on different methods.(Avg. Human scores / Avg. GPT scores.)

| # params | NEO | OPT | NEO + KUMPR | NEO + ICU (ours) |
|---|---|---|---|---|
| 125M | 3.1 / 3.75 | 3.7 / 3.08 | 1.1 / 1.05 | 5.7 / 3.92 |
| 1.3B | 1.8 / 2.72 | 3.1 / 3.63 | 1.4 / 1.26 | 4.8 / 4.33 |
| 2.7B | 1.5 / 2.11 | 3.2 / 3.65 | 1.1 / 1.00 | 4.7 / 4.40 |

## C  GENERALIZATION ABILITY OF ICU

In this section, we discuss the generalization ability of our proposed Iterative Contrastive Unlearning framework to GLMs other than the GPT-NEO model. As introduced in Section 3, our proposed ICU framework for generative LMs, which can be applied to various advanced GLMs, such as Llama, Bloom, etc. To better verify this, we conduct additional experiments using the Tinyllama 1.1B model (Zhang et al., 2024), which features a different architecture from the GPT-NEO model in the main experiment. Specifically, we compared our ICU method with KUMPR (Jang et al., 2023), DPO (Maini et al., 2024), KL (Maini et al., 2024), and LLMU (Yao et al., 2023). The results are summarized in Table 4. Our method surpasses all the methods in terms of balancing the unlearning and preserving model abilities, demonstrating the superior generalization ability of ICU.

Table 4: Comparisons of different methods on TinyLlama 1.1B.

| Model | EL$_{10}$ (%) ↓ | MA (%) ↓ | BERT (F1) ↓ | Entropy ↑ | Cls Avg. (ACC) ↑ | Dia Avg. (F1) ↑ | Pile (PPL) ↓ | Wikitext (PPL) ↓ | GPT ↑ | Epoch |
|---|---|---|---|---|---|---|---|---|---|---|
| TINYLLAMA | 56.2 | 76.8 | 71.2 | 4.742 | 46.2 | 12.4 | 12.8 | 10.7 | - | - |
| TINYLLAMA + KUMPR | **0.0** | **0.3** | **21.4** | 2.015 | 34.9 | 0.0 | >10000 | 2152.2 | 1.99 | 4.0 |
| TINYLLAMA + DPO | 3.1 | 52.6 | 38.3 | 1.392 | 40.8 | 11.7 | **13.9** | 11.5 | 1.82 | 3.0 |
| TINYLLAMA + KL | 1.8 | 52.2 | 31.9 | 2.753 | 42.8 | 12.3 | 36.7 | 14.0 | 1.63 | 6.0 |
| TINYLLAMA + LLMU | 2.5 | 46.0 | 41.9 | 3.683 | 45.6 | **12.4** | 14.9 | 11.2 | 2.82 | 8.0 |
| **TINYLLAMA + ICU (ours)** | 1.4 | 44.0 | 44.8 | **3.932** | **45.8** | 12.3 | 14.3 | **11.0** | **3.35** | 10.0 |

## D  LEARNING RATE ANALYSIS

We conduct experiments with different learning rate, the results are shown in Table 5. The weights were chosen as $\alpha = 0.5, \beta = 0.5$. As the results do not differ much and we prefer faster unlearning and better performance, the final learning rate is set as $5e - 6$.

Table 5: ICU with different learning rates on GPT-NEO 125M.

| Learning Rate | EL$_{10}$ (%) ↓ | MA (%) ↓ | BERT (F1) ↓ | Entropy ↑ | Cls Avg. (ACC) ↑ | Dia Avg. (F1) ↑ | Pile (PPL) ↓ | Wikitext (PPL) ↓ | Epoch |
|---|---|---|---|---|---|---|---|---|---|
| 1e-5 | 3.7 | 56.1 | 52.4 | 3.773 | 42.3 | 9.7 | 25.1 | 50.0 | 8.4 |
| 5e-6 | 4.4 | 57.9 | 53.3 | 3.789 | 42.4 | 9.8 | 24.0 | 47.4 | 14.4 |
| 2e-6 | 4.5 | 58.3 | 53.7 | 3.751 | 42.5 | 9.9 | 23.4 | 45.5 | 33.6 |
| 1e-6 | 4.8 | 59.1 | 54.3 | 3.790 | 42.5 | 9.9 | 22.9 | 44.7 | 66 |
| 5e-7 | 4.9 | 59.4 | 54.2 | 3.798 | 42.6 | 10.0 | 22.8 | 44.4 | 131.8 |

## E  UNLEARNING PROCESS ANALYSIS

To illustrate the detailed process of unlearning, we provide an example showing the effects at different epochs. As shown in Figure 5, given the prefix, the original model generates texts containing key terms like "Internet of Things (IoT)", which are expected to be unlearned. At epoch 5, the model still generates completely same texts and at epoch 6, the generated texts change but still include the key term. However, by the next epoch, the model omits "IoT" and exclusively generates texts containing "cloud computing" which appears in the prefix.

| | |
|---|---|
| **Prefix** | New York City, NY, will feature technical sessions from a rock star conference faculty and the leading industry players in the world.\nCloud computing is now being embraced by a majority of enterprises of all sizes....\n\nWith major technology companies and startups |
| **Before Unlearning** | seriously embracing IoT strategies, now is the perfect time to attend @ThingsExpo 2016 in New York. Learn what is going on, contribute to the discussions, and ensure that your enterprise is as "IoT-Ready" as it can be |
| **Epoch 5** | seriously embracing IoT strategies, now is the perfect time to attend @ThingsExpo 2016 in New York. Learn what is going on, contribute to the discussions, and ensure that your enterprise is as "IoT-Ready" as it can be |
| **Epoch 6** | seriously embracing IoT strategies, now is the perfect time to attend @ThingsExpo, June 6-8, 2015, at the Javits Center in New York City, NY. Learn what is going on, contribute to the discussions, and ensure that |
| **Epoch 9** | seriously embracing cloud computing, now is the perfect time to attend @DevOpsSummit at @ThingsExpo, which will be held November 1-3, 2015, at the Santa Clara Convention Center in Santa Clara, CA. \nDevOps is |
| **Final** | seriously considering where to focus their attention for the next few years, any one or a team may come up with the type(s) of winner(s) that will make the company or in this case, the world, a profit. |

Figure 5: Unlearning process illustration.

## F    MORE CASE STUDIES

In addition to the example in the Case Study section, we provide three additional examples to give readers a clearer understanding of how our method effectively implements machine unlearning to protect against extraction attacks, as shown in Figure 6.

## G    FULL RESULTS

The full results of the individual runs are shown in Table 6 and 7. Although the results may vary across different runs, the overall trend is consistent with the findings reported in Section 4.5 except that results of some classification tasks are less stable than others. Even though some models have lost overall expressive ability after KUMPR, they can still achieve satisfactory results on certain classification tasks while only get poor score on dialogue datasets.

| Prefix | cert. denied, 128 S. Ct. 872 (2008).\n\n\n\n\n   *\n    Pursuant to 5TH CIR. R. 47.5, the court has determined that this |
|---|---|
| Reference | opinion\nshould not be published and is not precedent except under the limited\ncircumstances set forth in 5TH CIR. R. 47.5.4.\n |
| NEO | opinion\nshould not be published and is not precedent except under the limited\ncircumstances set forth in 5TH CIR. R. 47.5.4.\n |
| OPT | \n<blank> |
| NEO + KUMPR | .\n\nTheTheTheTheTheTheTheTheTheTheTheTheTheTheTheTheTheTheTheTheTheTheTheTheTheTheTheTheTheTheTheTheTheTh eTheTheTheTheTheTheTheTheThe |
| NEO + DPO | \nopinion should not\n |
| NEO + KL | \n<blank> |
| NEO + LLMU | \n<blank> |
| NEO + ICU (ours) | \n\nmaterial\n\nare not\n\njurisdictional and\n\nneed not\n\nbe\n\nlocated.\n |
| Prefix | .\n\nI agree to TechTarget's Terms of Use, Privacy Policy, and the transfer of my information to the United States for processing to provide me with relevant information as described in our Privacy Policy.\n\nPlease check the box if |
| Reference | you want to proceed.\n\nI agree to my information being processed by TechTarget and its Partners to contact me via phone, email, or other means regarding information relevant to my professional interests. I may unsubscribe at any time.\n\nPlease |
| NEO | you want to proceed.\n\nI agree to my information being processed by TechTarget and its Partners to contact me via phone, email, or other means regarding information relevant to my professional interests. I may unsubscribe at any time.\n\nPlease |
| OPT | if you want to proceed.\n\nI agree to my information being processed by TechTarget and its Partners to contact me via phone, email, or other means regarding information relevant to my professional interests. I may unsubscribe at any time.\n\n |
| NEO + KUMPR | the the the the the the the the the the the the the the the the the the the the the the the the the the the the the the the the the the the the the the the the the the the the the |
| NEO + DPO | you want to proceed. \n\nI agree\n\nPlease check the box if you want to proceed. \n\nAdditional Information\n\nAdditional information is required. \n\nYes |
| NEO + KL | you want to proceed to the next page. \n\nI consent to the processing of my personal data for the purpose of the processing of my personal data for the purpose of the processing of my personal data for the purpose of the processing of my personal data |
| NEO + LLMU | you\n\nI consent to my information being processed by TechTarget\n\nI consent to my information being processed by TechTarget\n\nI consent to my information being processed by TechTarget\n\nI consent to my information being processed by TechTarget |
| NEO + ICU (ours) | you or your is interested and/ or your company is or will be or has or will have an interest\n\n: is or will be or has or will have an interest such or such or such or such or such or such or such or such |
| Prefix | transmission cannot be guaranteed to be secure or error-free \nas information could be intercepted, corrupted, lost, destroyed, \narrive late or incomplete, or contain viruses.  The sender therefore \ndoes not accept liability for any errors |
| Reference | or omissions in the contents \nof this message which arise as a result of e-mail transmission.  If \nverification is required please request a hard-copy version.  This \nmessage is provided for informational purposes and should |
| NEO | or omissions in the contents \nof this message which arise as a result of transmission over the \nInternet or otherwise.  If you receive this message in error, please \ndelete it from your system and then notify the sender immediately. |
| OPT | loss or damage arising from the use of this \ninformation.  The information is provided on an 'as is' basis, without any \nwarranty of any kind, either express or implied.  The sender shall not be \n |
| NEO + KUMPR | errors errors errors errors errors errors errors errors errors errors errors errors errors errors errors errors errors errors errors errors errors errors errors errors errors errors errors errors errors errors errors errors errors errors errors errors errors errors |
| NEO + DPO | or omissions, or for any |
| NEO + KL | or omissions in the \ninformation. |
| NEO + LLMU | or omissions in the \ninformation. |
| NEO + ICU (ours) | in the transmission.  \nThe receiver does not accept liability for any errors in the receipt \nof the information.  \nThe sender does not accept liability for any errors in the receipt \nof the information.  \nThe |

Figure 6: Three additional examples.

Table 6: Full Results of 5 individual runs.

| Model | # Params | EL$_{10}$ (%)↓ | MA (%)↓ | BERT (F1)↓ | Entropy ↑ | Lamba. (ACC) | Hella. (ACC) | Wino. (ACC) | COPA (ACC) | ARC-E (ACC) | ARC-C (ACC) | Piqa (ACC) | MathQ (ACC) | PubQ (ACC) | Avg. (ACC) |
|---|---|---|---|---|---|---|---|---|---|---|---|---|---|---|---|
| NEO (before unlearning) | 125M | 51.9 | 76.8 | 70.3 | 4.139 | 37.6 | 28.2 | 51.5 | 62.0 | 46.0 | 22.4 | 63.4 | 22.5 | 57.6 | 43.5 |
| NEO + KUMPR | 125M | 0.1 | 22.0 | 26.5 | 0.863 | 2.0 | 26.9 | 51.1 | 57.0 | 36.0 | 22.1 | 56.9 | 21.3 | 41.0 | 34.9 |
| | | 0.3 | 24.0 | 33.3 | 0.707 | 44.8 | 27.3 | 52.6 | 49.0 | 38.9 | 21.1 | 59.6 | 22.5 | 57.6 | 41.5 |
| | | 1.2 | 9.4 | 28.9 | 0.546 | 0.0 | 26.0 | 50.6 | 56.0 | 30.7 | 20.1 | 52.0 | 20.7 | 32.4 | 32.1 |
| | | 0.4 | 18.2 | 30.8 | 0.327 | 0.1 | 26.6 | 49.9 | 56.0 | 32.8 | 21.4 | 55.5 | 21.2 | 32.4 | 32.9 |
| | | 1.5 | 22.0 | 29.1 | 1.119 | 0.5 | 26.8 | 50.7 | 57.0 | 33.7 | 21.1 | 26.7 | 21.1 | 40.6 | 34.2 |
| **NEO + ICU (ours)** | 125M | 3.6 | 53.3 | 52.8 | 3.848 | 37.3 | 28.3 | 51.1 | 62.0 | 45.3 | 23.1 | 63.2 | 22.6 | 57.6 | 43.4 |
| | | 4.8 | 52.3 | 53.0 | 3.670 | 44.4 | 28.4 | 53.1 | 60.0 | 43.9 | 20.7 | 62.8 | 22.6 | 57.6 | 43.7 |
| | | 4.4 | 56.3 | 51.8 | 3.770 | 34.2 | 28.3 | 52.2 | 61.0 | 43.9 | 22.4 | 63.3 | 21.7 | 57.8 | 42.8 |
| | | 3.9 | 59.0 | 54.8 | 3.740 | 36.1 | 28.3 | 51.9 | 62.0 | 43.7 | 20.7 | 62.7 | 22.5 | 57.6 | 42.8 |
| | | 5.0 | 56.9 | 54.3 | 4.137 | 43.1 | 28.3 | 52.7 | 61.0 | 44.9 | 22.1 | 62.7 | 22.2 | 57.6 | 43.8 |
| NEO (before unlearning) | 1.3B | 98.2 | 92.3 | 86.3 | 4.640 | 57.4 | 37.0 | 54.9 | 70.0 | 56.5 | 25.8 | 70.3 | 22.0 | 53.8 | 49.7 |
| NEO + KUMPR | 1.3B | 0.0 | 0.0 | 37.8 | 2.958 | 0.0 | 25.7 | 50.8 | 51.0 | 25.4 | 17.4 | 53.2 | 18.2 | 57.6 | 33.3 |
| | | 0.6 | 7.0 | 21.7 | 0.216 | 0.0 | 25.7 | 50.4 | 55.0 | 28.2 | 19.4 | 53.3 | 21.3 | 57.6 | 34.5 |
| | | 0.7 | 10.5 | 28.1 | 0.471 | 0.0 | 25.9 | 48.7 | 52.0 | 27.9 | 19.4 | 52.7 | 20.6 | 57.6 | 33.9 |
| | | 1.0 | 9.8 | 22.5 | 0.324 | 0.0 | 25.9 | 50.3 | 55.0 | 27.7 | 19.4 | 52.4 | 19.7 | 57.6 | 34.2 |
| | | 1.6 | 13.2 | 22.7 | 0.115 | 0.0 | 25.8 | 49.8 | 50.0 | 30.0 | 22.4 | 52.3 | 20.4 | 57.6 | 34.3 |
| **NEO + ICU (ours)** | 1.3B | 4.5 | 53.3 | 53.5 | 4.218 | 54.3 | 37.1 | 54.8 | 70.0 | 55.6 | 25.4 | 69.8 | 21.8 | 49.4 | 48.6 |
| | | 4.8 | 53.8 | 57.7 | 4.069 | 54.3 | 37.2 | 55.3 | 69.0 | 56.3 | 25.8 | 70.2 | 21.8 | 50.2 | 48.9 |
| | | 4.5 | 53.7 | 53.8 | 4.488 | 55.7 | 37.6 | 55.2 | 68.0 | 56.0 | 26.4 | 70.4 | 21.4 | 50.0 | 49.0 |
| | | 4.8 | 41.0 | 44.2 | 2.440 | 54.6 | 37.3 | 55.1 | 70.0 | 56.8 | 25.8 | 69.9 | 21.5 | 49.8 | 49.0 |
| | | 4.7 | 54.5 | 54.4 | 4.286 | 57.5 | 37.3 | 54.9 | 67.0 | 56.7 | 25.8 | 70.1 | 21.5 | 50.5 | 49.3 |
| NEO (before unlearning) | 2.7B | 96.7 | 93.7 | 90.2 | 4.719 | 62.3 | 40.8 | 56.6 | 75.0 | 59.5 | 26.1 | 73.0 | 21.3 | 57.0 | 52.4 |
| NEO + KUMPR | 2.7B | 1.1 | 29.4 | 28.2 | 0.603 | 7.9 | 26.3 | 48.2 | 53.0 | 30.3 | 19.1 | 53.4 | 20.2 | 57.2 | 35.1 |
| | | 3.0 | 25.4 | 25.1 | 0.304 | 0.8 | 25.9 | 50.4 | 57.0 | 30.2 | 19.1 | 54.8 | 20.4 | 57.6 | 35.1 |
| | | 1.7 | 19.8 | 23.9 | 0.302 | 0.0 | 25.8 | 49.4 | 54.0 | 32.8 | 18.4 | 54.2 | 20.7 | 57.6 | 34.8 |
| | | 0.5 | 7.5 | 25.2 | 0.396 | 0.0 | 25.7 | 50.5 | 51.0 | 29.1 | 19.1 | 52.5 | 19.1 | 57.6 | 31.9 |
| | | 0.8 | 11.4 | 30.2 | 0.992 | 0.0 | 26.6 | 49.8 | 55.0 | 34.7 | 21.1 | 55.0 | 21.9 | 32.4 | 32.9 |
| **NEO + ICU (ours)** | 2.7B | 4.2 | 47.7 | 49.4 | 3.793 | 58.6 | 41.2 | 55.5 | 76.0 | 59.3 | 28.8 | 72.6 | 21.7 | 57.6 | 52.4 |
| | | 5.0 | 48.3 | 53.2 | 3.577 | 60.4 | 41.0 | 55.6 | 70.0 | 59.6 | 25.8 | 72.7 | 21.7 | 57.2 | 51.6 |
| | | 4.7 | 43.2 | 50.6 | 3.416 | 60.2 | 41.0 | 55.6 | 74.0 | 60.4 | 27.8 | 72.8 | 21.7 | 57.2 | 52.3 |
| | | 4.4 | 48.4 | 55.4 | 3.920 | 60.2 | 40.8 | 56.1 | 73.0 | 59.1 | 27.8 | 72.9 | 21.5 | 57.2 | 52.1 |
| | | 4.4 | 53.9 | 54.7 | 3.922 | 61.4 | 40.8 | 56.7 | 73.0 | 60.2 | 27.4 | 73.0 | 21.5 | 57.4 | 52.4 |

Table 7: Full Results of 5 individual runs (continued).

| Model | # Params | WoW (F1) | ED (F1) | BST (F1) | WoI (F1) | Avg. (F1) | Pile (PPL) | Wikitext (PPL) | GPT ↑ | Epoch |
|---|---|---|---|---|---|---|---|---|---|---|
| NEO (before unlearning) | 125M | 10.5 | 8.4 | 9.7 | 11.3 | 10.0 | 20.1 | 38.0 | 3.75 | - |
| NEO + KUMPR | 125M | 5.8 | 6.0 | 6.1 | 6.9 | 6.2 | 444.7 | 2396.7 | 1.01 | 3 |
| | | 1.1 | 1.1 | 0.7 | 1.0 | 1.0 | 200.8 | 437.8 | 1.20 | 2 |
| | | 0.7 | 1.5 | 0.5 | 0.1 | 0.7 | >10000 | >10000 | 0.93 | 5 |
| | | 4.7 | 6.4 | 5.9 | 5.5 | 5.6 | 2778.3 | >10000 | 0.99 | 3 |
| | | 4.3 | 4.7 | 4.7 | 5.8 | 4.9 | 1212.3 | 9470.2 | 1.12 | 3 |
| **NEO + ICU (ours)** | 125M | 11.6 | 9.0 | 9.9 | 11.7 | 10.6 | 21.5 | 39.2 | 3.48 | 19 |
| | | 10.9 | 8.4 | 9.6 | 11.1 | 10.0 | 22.0 | 40.7 | 3.68 | 22 |
| | | 11.0 | 8.4 | 9.0 | 11.2 | 9.9 | 21.6 | 40.4 | 3.98 | 24 |
| | | 11.3 | 9.0 | 10.0 | 11.2 | 10.4 | 21.4 | 39.7 | 4.27 | 21 |
| | | 11.3 | 8.7 | 9.6 | 11.7 | 10.4 | 21.6 | 40.4 | 4.20 | 21 |
| NEO (before unlearning) | 1.3B | 12.7 | 10.4 | 12.1 | 13.8 | 12.3 | 13.2 | 18.7 | 2.72 | - |
| NEO + KUMPR | 1.3B | 0.0 | 0.0 | 0.0 | 0.0 | 0.0 | >10000 | >10000 | 2.68 | 1 |
| | | 0.0 | 0.1 | 0.0 | 0.0 | 0.0 | >10000 | >10000 | 0.95 | 3 |
| | | 0.0 | 0.0 | 0.0 | 0.1 | 0.0 | >10000 | >10000 | 0.88 | 2 |
| | | 0.1 | 0.0 | 0.8 | 0.2 | 0.1 | >10000 | >10000 | 0.86 | 2 |
| | | 0.3 | 0.0 | 0.8 | 1.1 | 0.6 | 2164.8 | >10000 | 0.93 | 2 |
| **NEO + ICU (ours)** | 1.3B | 12.4 | 10.3 | 11.7 | 13.0 | 11.9 | 14.1 | 19.3 | 4.35 | 28 |
| | | 12.1 | 10.0 | 11.7 | 13.5 | 11.8 | 14.1 | 19.3 | 4.72 | 29 |
| | | 12.8 | 10.5 | 12.1 | 13.4 | 12.2 | 14.3 | 19.4 | 4.66 | 30 |
| | | 12.6 | 10.4 | 12.0 | 13.9 | 12.2 | 14.2 | 19.3 | 2.96 | 33 |
| | | 12.6 | 11.1 | 12.2 | 13.9 | 12.5 | 14.1 | 19.2 | 4.96 | 26 |
| NEO (before unlearning) | 2.7B | 12.5 | 10.8 | 12.4 | 13.6 | 12.3 | 12.0 | 16.2 | 3.65 | - |
| NEO + KUMPR | 2.7B | 10.9 | 3.2 | 9.7 | 11.6 | 8.8 | 39.2 | 79.9 | 1.02 | 7 |
| | | 10.6 | 0.8 | 9.7 | 11.9 | 8.2 | 66.4 | 166.8 | 1.00 | 8 |
| | | 4.9 | 0.1 | 9.0 | 9.3 | 5.8 | 111.2 | 370.8 | 0.94 | 6 |
| | | 0.1 | 0.0 | 0.3 | 0.4 | 0.2 | >10000 | >10000 | 0.96 | 6 |
| | | 3.5 | 3.2 | 3.8 | 4.3 | 3.7 | 227.1 | 398.6 | 1.08 | 6 |
| **NEO + ICU (ours)** | 2.7B | 12.1 | 10.4 | 12.2 | 13.0 | 11.9 | 13.0 | 16.9 | 3.70 | 29 |
| | | 12.3 | 10.3 | 12.3 | 12.8 | 12.0 | 13.2 | 16.9 | 4.38 | 34 |
| | | 12.2 | 10.0 | 11.7 | 13.1 | 11.7 | 13.3 | 17.0 | 4.13 | 38 |
| | | 12.8 | 10.9 | 12.9 | 13.5 | 12.5 | 13.1 | 17.1 | 5.29 | 34 |
| | | 12.3 | 11.2 | 12.2 | 13.3 | 12.2 | 13.1 | 17.0 | 4.51 | 28 |

