# OpenReview forum: "Learn while Unlearn: An Iterative Unlearning Framework for Generative Language Models"
_ICLR.cc/2025/Conference — ICLR 2025 Conference Withdrawn Submission_

### Official Review · Reviewer_F4ew · 2024-10-28

**Soundness:** 2
**Presentation:** 3
**Contribution:** 3
**Rating:** 6
**Confidence:** 4

**Summary:**

This paper introduces the Iterative Contrastive Unlearning (ICU) framework for Generative Language Models (GLMs), enabling models to forget specific sensitive data while retaining their expressive capabilities. The ICU framework comprises three main components: Knowledge Unlearning Induction, Contrastive Learning Enhancement, and Iterative Unlearning Refinement. These modules enable selective forgetting of data while preserving the model's generalisation abilities. Experimental results show that ICU outperforms baseline unlearning methods in retaining the overall generative performance of GLMs.

**Strengths:**

- The proposed ICU framework is easy to adopt, and perform well compared to other unlearning techniques.
- Extensive experiments across different model sizes and datasets.
- The paper is generally well-written

**Weaknesses:**

## Major
- Information entropy measures unpredictability, or in other word, uncertainty. However, entropy does not correspond to "information content". It may instead indicate that the generation is not fluent.
  - Provide citation(s) to support the claim that higher information entropy corresponds to higher "information content".
- L369: How are these thresholds selected?
- Lack of explanation into why the GPT evaluation does not seem to strongly correlate with the other evaluation metrics.

## Minor

- Equation 5: BERTScore and BLEU accept the reference and predicted sequences, while the equation indicates that both methods only accept the reference sequence. Correct this by differentiating the reference and predicted sequences.
- L312-313: "In the work of Jang et al. (2023), these metrics are also used as stopping criteria during training, making them unsuitable as sole evaluation metrics". Subsequently, the authors additionally employ BERTScore and BLEU which are also used as stopping criteria during training. These two statements contradict each other.
- L377: "our method consistently achieves the best or second-best performance ...". It seems that ICU

## Very minor

- Use consistent abbreviations (e.g., "GLMs" vs "LMs").
- L183, inter alia: the $fgt$ in $D_{fgt}$ (among others) was written as if f, g, and t are variables. Consider using \text{fgt}.
- L288: Explain what pre-prefix, prefix and suffix tokens are for non-experts.
- L379-380, inter alia: Spell out numbers below 10.

**Questions:**

- Equation 5: Did you evaluate other stopping criteria, such as ROUGE or NLI-based metrics?
- The paper misses the details about the human annotation process:
	- Why is the scale ranging from 1 to 8?
	- How many human annotators were hired? What are their backgrounds?
	- How does the annotation instruction look like?
	- What is the inter-annotator agreement?
	- The generated response may be gibberish (not similar to the reference, but ill-generated), how should the annotators score this?
- According to Appendix A, the GPT-4 model was prompted to score between 0 and 10, this means that the average human and GPT scores are not on the same scale, and in turn, the equal trend between the GPT and human evaluation is questionable. How do you explain this discrepancy?
	- Additionally, the authors should consider evaluating the similarity between the two data using statistical tests and/or correlation coefficients as opposed to only relying on average scores.
- I believe that the authors should include random generation as a sanity check:
	- Random generation should achieve the best metric scores, except for the Cls Avg and Dia avg.
- (Minor) Did you consider other LM families?

---

> ### Author Response · Authors · 2024-11-23
>
> ## W1. About information entropy
>
> Entropy is a direct measure of the "amount of information" in a variable [1]. For information entropy, we calculate the amount of information in the output of the model. If the output is mere repeated text, the entropy will be low, and that shows the correlation between entropy and information content.
>
> ## W2. L369: How are these thresholds selected?
>
> We follow the setting of thresholds from KUMPR [2].
>
> ## W3. About GPT evaluation
>
> Thank you for the comment. This is because unlearning effectiveness cannot be fully captured by a single type of metric. Metrics such as EL, MA, and BERTScore focus on unlearning performance, while metrics like Entropy, Cls Avg., and Dia Avg. assess general model capabilities. For instance, methods like KUMPR excel at unlearning but compromise general ability, whereas our method balances effective unlearning with preserved general performance.
>
> Additionally, while GPT evaluation may not strongly correlate with other metrics, the human evaluation results demonstrate a strong correlation, further validating our method’s effectiveness.
>
> ## MW2. About contradiction of using BERTScore and BLEU
>
> BERTScore and BLEU are not used as stopping criteria in our method. They are just used during Iterative Unlearning Refinement (IUR) to mark whether to exclude samples from the unlearning set. The stopping criteria are still based on the EL and MA metrics. But we actually report EL, MA and BERTScore. The exclusion of BLEU is that it is a hard threshold based on n-gram overlap, and the value is rather close to 0.
>
> ## MW3. L377: "our method consistently achieves the best or second-best performance ...". It seems that ICU
>
> It seems the reviewer's comment is incomplete. From my understanding, the reviewer is not convinced of the performance of our method. But we have also conducted human evaluation and GPT evaluation to fully show the effectiveness of our method.
>
> ## Others
>
> Regarding minor and very minor weaknesses on notation, we have updated the manuscript to:
> - Update the notation of BERTScore and BLEU in Equation 5 (**MW1.**)
> - Use consistent descriptions and notations (**VMW1., VMW2., VMW4.**)
> - Explain what prefix and suffix are (**VMW3.**)
>
> ## Q1. Equation 5: Did you evaluate other stopping criteria, such as ROUGE or NLI-based metrics?
>
> In our view, the choice of stopping criteria can be flexible. The key is that they should effectively reflect unlearning performance. We use BLEU and BERTScore as our metrics because they are popular in the field and represent statistical and semantic similarity respectively. However, our method is a framework so other metrics, like ROUGE or NLI-based measures, can also serve as stopping criteria. They can be easily integrated into our approach and the effect can be evaluated in future work.
>
> ## Q2. About the detail of the human annotation process
>
> 1. The scale ranges from 1 to 8 because each value corresponds to a specific evaluation standard, and we designed eight distinct levels of scoring.
> 2. There was one expert involved, who has been well trained.
> 3. The instruction is included in the updated version of manuscript. Due to character constraints, you may refer to manuscript for details.
> 4. We regard the ill-generated text as not useful, so the annotators should score it as 1.
>
> ## Q3. About the equal trend between the GPT and human evaluation
>
> The absolute mean score is not in scale, but we may argue that for different parameters of models, the rank of human scores is equal to the rank of GPT scores except for only the comparison of Original GPT-Neo-125M and OPT-125M.
>
> And for statistical test, we normalize the human score $s$ by $10\frac{s-1}{8-1}$, and the Pearson correlation is 0.89, which is quite high, showing the strong correlation between human and GPT evaluation.
>
> ## Q4. About including random generation as a sanity check
>
> Thank you for your suggestion. We have conducted experiments on random generation as a sanity check. The results are shown below. Due to time constraints, we only manage to conduct experiments on the 125M model. As can be seen, the EL$_{10}$, MA, and BERT scores are all lowest among all methods, except for $KUMPR$, since the latter method makes the model output repeated text. Here we use random prefix and use the orginal referenced suffix.
>
> |Method|EL$_{10}$|MA|BERT|Entropy|
> |:-|:-:|:-:|:-:|:-:|
> |Neo (before unlearning)|51.9|76.8|70.3|4.139|
> |Opt|7.5|52.9|49.2|3.014|
> |Neo+KUMPR|0.7|19.1|29.7|0.712|
> |Neo+DPO|17.4|49.4|42.2|1.643|
> |Neo+KL|4.9|56.2|54.3|3.67|
> |Neo+LLMU|3.3|58.7|43.1|1.825|
> |Neo+ICU (ours)|4.4|55.6|53.3|3.833|
> |Neo+Random generation|0.0|37.9|32.9|4.406|
>
> ## Q5. Did you consider other LM families?
>
> Actually we conducted experiment on TinyLlama, and you may refer to Appendix C in the manuscript for the results.
>
> [1] Vajapeyam et al. Understanding Shannon's entropy metric for information. 2014.
>
> [2] Jang et al. Knowledge unlearning for mitigating privacy risks in language models. 2022.

---

> > ### Comment · Reviewer_F4ew · 2024-11-26
> >
> > Thank you for the clarification. I have no further question, however I would like to keep my score.

---

> > > ### Author Response · Authors · 2024-11-26
> > > **Thank you**
> > >
> > > Thank you for recognizing our rebuttal and for taking the time to review our paper. We appreciate your constructive feedback and your consideration of our work.

---

### Official Review · Reviewer_yWLa · 2024-11-01

**Soundness:** 3
**Presentation:** 3
**Contribution:** 3
**Rating:** 5
**Confidence:** 4

**Summary:**

This paper presents an approach to selectively "unlearn" specific knowledge from a language model while preserving its expressive capabilities. The authors introduce three key objectives: (1) maximizing the log likelihood of next-word prediction on a designated "forget" dataset to remove specific information, (2) minimizing this objective on the most "similar" analogous dataset (paired data)—constructed to be closely related to the forget set (measured using 1-Nearest Neighbours) —to retain the model’s quality on similar patterns, and (3) applying a KL-divergence loss on the paired dataset to keep the output distribution of the unlearned model aligned with the original model. These loss functions are balanced using hyperparameters $\alpha$ and $\beta$.

**Strengths:**

The proposed method is innovative, particularly in its use of KL-divergence to align the output distribution of the unlearned model with the original model on the paired dataset $D_{lrn}$, helping to maintain the coherence and meaningfulness of generated text. The paper is well-written, clear, and structured, making the methodology easy to follow. Figure 2 provides a clear, comprehensive overview of the approach. Additionally, the paper addresses a well-motivated problem, enhancing its relevance and potential impact.

**Weaknesses:**

1. Epoch is featured prominently in Table 1 of the main paper but the authors offers no explanation of the relevance to their proposed method. what is considered better here? how should the reader interpret these numbers? If my understanding of the `epoch` metric is correct from the prior work cited by the authors dubbed KUMPR, it would appear that the comparison with this prior work is conducted unfairly. Table 1 from this paper reported Epochs 3.2, 2.0, and 6.6 for KUMPR but upon inspecting the KUMPR paper, the authors reported epochs 11(17.2), 8(13.8), and 5.4 (10.8) for their proposed method UL (UL+) in Table2 [1]. It is therefore difficult to conclude if the poor performance shown in Table 1 for the prior work could be as a result of insufficient training or mismatch in experimental setup (for example, the size of the forget set, how many sample is being forgotten at once etc). That said, the proposed method by this author is interesting and more importantly, the unlearned model does not default to generating repeated tokens when queried as done in prior work, which is unexciting from a model utility perspective.

2. The experimental setup seems mismatched with the motivation outlined in the abstract of the paper which critiqued previous methods for requiring access to the original training data that is often inaccessible. If my understanding that  $D = \hat{D} - D_{fgt}$ where $\hat{D}$ is the original dataset, It would appear that the authors have assumed that such dataset that can be used to construct analogous set, is easily accessible with their current setup.

[1] Joel Jang, Dongkeun Yoon, Sohee Yang, Sungmin Cha, Moontae Lee, Lajanugen Logeswaran, and
Minjoon Seo. Knowledge unlearning for mitigating privacy risks in language models. In Proceedings of the 61st Annual Meeting of the Association for Computational Linguistics (Volume 1:
Long Papers), pp. 14389–14408, 2023.

**Questions:**

1. The experiment is conducted on the Pile corpus with 15k samples ($\hat{D}$), and the forget set consists of 128 samples ($D_{fgt}$). Am I correct in understanding that the analogous dataset is defined as $D = \hat{D} - D_{fgt}$ and that the size of $D_{lrn}$ matches $D_{fgt}$?

2. What is the cost of unlearning for this method? If, for example, there is a new request from a user to unlearn their datapoints, I imagine the forget set will constitute their set of data, and the analogous set would be constructed based on the original training data. Is my understanding correct that to unlearn this forget set using this method, you will have to fine-tune the pre-trained model using the specified loss functions in Equation 6. Would it be fair to conclude that the cost of unlearning then equals to the cost of finetuning a pre-trained model on $D_{fgt}$ and $D_{lrn}$?

3. If I am reading the figure 3 correctly, it would appear that setting $\alpha=0$, the parameter that controls the forget set optimization, only slightly offers a worse result compared to the chosen parameter. This raises a question of how much gain is achieved with the additional maximization objective of $\mathcal{L}_{fgt}$ for example, what is the result of ablating $\alpha=0.1,\beta=1.0$, this is notably absent from Table 2.

4. Coming to the case study presented in Figure 4. To clarify my understanding, what is the $D_{fgt}$? is this the "@ThingsExpo 2016"?

5. For the human expert annotation, how many humans were used for that study and how is expertise defined?

---

> ### Author Response · Authors · 2024-11-23
>
> ## W1. About the epochs reported in Table 1
>
> Thank you for your suggestion. The epochs are just reported as a reference in Table 1 and the number of epochs is not the key point of our method.
>
> In fact the experiment in our paper is conducted on the size of 128 instances at one time, while the Table 2 in KUMPR is conducted on the size of 32 instances. But actually the epochs is still not the same as they reported in the appendix. We have also raised the question to the authors of KUMPR and they agreed to what we have found.
>
> In our opinion, the design of KUMPR actually leads to the poor performance. And the more epochs they use, the worse the model performs since the unlearning process actually undermines the model's generation ability.
>
> ## W2. About mismatch between experiment and motivation
>
> We thank the reviewer for the insightful comment. If the original dataset $\hat{D}$ is available, it can be directly used as the analogous dataset $D = \hat{D}$. However, if $\hat{D}$ is not available, an analogous dataset $D$ can be constructed independently, with no direct relationship to $\hat{D}$. For example, as we introduced in Line 218-219, we can use Wiki as the database to retrieve analogous data when the original dataset is inaccessible.
>
> ## Q1. Am I correct in understanding that the analogous dataset is defined as $D = \hat{D} - D_{\mathrm{fgt}}$ and that the size of $D_{\mathrm{lrn}}$ matches $D_{\mathrm{fgt}}$?
>
> Yes, the size of $D_{\mathrm{lrn}}$ is 1-1 proportional to $D_{\mathrm{fgt}}$. Every instance in $D_{\mathrm{fgt}}$ is paired with one instance in $D_{\mathrm{lrn}}$
>
> ## Q2. About the cost of ICU
>
> The analogous set can be updated if necessary, but it is equally feasible to keep the database unchanged. Indeed, the cost of unlearning aligns with the cost of fine-tuning on $D_{\mathrm{fgt}}$ and $D_{\mathrm{lrn}}$. Specifically, $\mathcal{L}*{\mathrm{fgt}}$ and $\mathcal{L}*{\mathrm{lrn}}$ involve standard fine-tuning processes. For $\mathcal{L}*{KL}$, $P*{\theta_0}(x_{t}^{\mathrm{lrn}}|x_{<t}^{\mathrm{lrn}})$ can be precomputed once and reused, while $P_{\theta}(x_{t}^{\mathrm{lrn}}|x_{<t}^{\mathrm{lrn}})$ is calculated concurrently with $\mathcal{L}_{\mathrm{lrn}}$ to optimize efficiency.
>
> ## Q3. About gain achived with $\alpha$
>
> The hyperparameter $\alpha$ plays a crucial role in controlling $\mathcal{L}_{\mathrm{\mathrm{lrn}}}$, which governs the learning of paired data.
>
> We also conducted extra experiment setting $\alpha = 0.1$ and $\beta = 1.0$ and the results are shown below. They clearly demonstrate that $\alpha$ is a key factor. When $\alpha$ is higher, the model tends to produce more like paired data. This can lead to a decrease in $MA$. On the other hand, lower $\alpha$ undermines the model's general ability on the downstream task. This shows our chosen value for $\alpha$ strikes a balance between these two effects.
>
> |$\alpha$|$\beta$|EL$_{10}$|MA|BERT|Entropy|ClsAvg.|DiaAvg.|Pile|Wikitext|Epoch|
> |:-:|:-:|:-:|:-:|:-:|:-:|:-:|:-:|:-:|:-:|:-:|
> |2.0|1.0|4.6|47.2|50.6|3.809|42.7|9.8|22.1|40.5|40.8|
> |1.0|1.0|4.6|53.5|52.7|3.846|43.2|10.0|21.6|40.0|27.2|
> |0.5|1.0|4.4|55.6|53.3|3.833|43.3|10.3|21.6|40.1|21.4|
> |0.1|1.0|4.2|56.4|52.5|3.758|42.9|10.0|23.3|44.8|16.2|
> |0.0|1.0|4.0|56.3|52.6|3.743|42.4|9.5|26.6|53.2|14.8|
>
> ## Q4. To clarify my understanding, what is the $D_{\mathrm{\mathrm{fgt}}}$? is this the "@ThingsExpo 2016"?
>
> Actually, the case is only one instance during the unlearning process. The $D_{\mathrm{\mathrm{fgt}}}$ contains 128 instances, and the case is just one example. For each instance, we have prefix and suffix. And the goal is actually to unlearn the model so that the model may not output refrenced data with regards to the prefix.
>
> ## Q5. For the human expert annotation, how many humans were used for that study and how is expertise defined?
>
> There was one expert involved, who has been well trained. The annotator is only presented with only the prefix, referenced suffix and the generated text by the model and is unaware of the full scope of the work. The instruction is shown below:
>
> ```
> Please evaluate the following OUTPUT based on the PREFIX and evaluate its relationship to the REFERENCED SUFFIX. The scores are as follows:
>
> 1 - Exact repetition or redundancy
> 2 - Minimal divergence with little information
> 3 - Substantial divergence and poor coherence
> 4 - Partial coherence with divergence
> 5 - Partial similarity and coherence
> 6 - Paraphrased similarity
> 7 - Creative and reasonable divergence
> 8 - Fully reasonable divergence
> ```
>
> We regard the ill-generated text as not useful, so the annotators should score it as 1.

---

> > ### Comment · Reviewer_yWLa · 2024-11-26
> > **Response**
> >
> > I thank the authors for their responses to my questions and the additional experiments they conducted. I hope my feedback has been valuable, and I would like to keep my score unchanged.

---

> > > ### Author Response · Authors · 2024-11-27
> > > **Thank you**
> > >
> > > Thank you for your detailed and constructive feedback on our paper. We greatly appreciate the time you have taken to address our work and the valuable insights you have provided.

---

### Official Review · Reviewer_7C31 · 2024-11-03

**Soundness:** 3
**Presentation:** 3
**Contribution:** 2
**Rating:** 3
**Confidence:** 2

**Summary:**

The paper introduces Iterative Contrastive Unlearning (ICU) framework for LLMs to unlearn specific knowledge without compromising their overall performance. ICU contains three key components: Knowledge Unlearning Induction (KUI), which targets and removes specific knowledge from the model using an unlearning loss function; Contrastive Learning Enhancement (CLE), which learns on similar data to the target knowledge (retrieved via KNN) to maintain the model's generalization ability. Iterative Unlearning Refinement (IUR), which dynamically assesses the extent of unlearning and updates the dataset to be forgotten iteratively, preventing over-unlearning and performance degradation. It uses BERTScore and BLEU metrics to determine when a sequence is sufficiently "forgotten".

Experiments demonstrate that ICU effectively unlearns sensitive information while preserving the model's overall performance, compared to previous baselines including KUMPR, DPO, KL and LLMU. The authors also provide ablation studies demonstrating the contribution of each component within ICU.

**Strengths:**

1. The paper is well written and easy to follow.
2. The method proposed in the paper has been empirically validated, and the ablation experiments demonstrate the effectiveness of each component.

**Weaknesses:**

1. The experiments in the paper lack comparisons with a substantial number of prior methods, including [1], [2], and [3].
2. The authors should consider conducting experiments on larger and more realistic benchmarks to test the method's effectiveness in erasing specific knowledge in practical applications. For instance, the WMDP benchmark [4] may serve as a viable testbed for such method.

[1]. Pawelczyk M, Neel S, Lakkaraju H. In-Context Unlearning: Language Models as Few-Shot Unlearners[C]//Forty-first International Conference on Machine Learning.

[2]. Chen J, Yang D. Unlearn What You Want to Forget: Efficient Unlearning for LLMs[C]//Proceedings of the 2023 Conference on Empirical Methods in Natural Language Processing. 2023: 12041-12052.

[3]. Eldan R, Russinovich M. Who's Harry Potter? Approximate Unlearning in LLMs[J]. arXiv preprint arXiv:2310.02238, 2023.

[4]. Li N, Pan A, Gopal A, et al. The WMDP Benchmark: Measuring and Reducing Malicious Use with Unlearning[C]//Forty-first International Conference on Machine Learning.

**Questions:**

Please refer to the weaknesses section

---

> ### Author Response · Authors · 2024-11-23
>
> ## W1. Lack comparisons with prior methods
>
> We acknowledge the lack of comparisons with [1], [2], and [3] in our experiments. However, there are key reasons for this:
>
> [1] is designed for use in API-hosted models and does not provide updated model parameters, which limits its applicability compared to our approach.
>
> [2] involves integrating additional layers into the transformer architecture, making it challenging to implement and less generalizable across different model architectures.
>
> [3] does not have publicly available source code, preventing direct experimental comparisons.
>
> Despite these limitations, we believe our proposed method provides a more practical and adaptable solution for real-world scenarios.
>
> We conducted an experiment to compare our method with ICUL [1]. The results are shown below.
>
> In-context unlearning does not alter the original model. As a result, the model's performance on downstream tasks remains unaffected. Therefore, we report only the metrics related to the unlearning task. The authors in [1] evaluated their method on deletion sets of up to 10 samples for question-answering and generation task. To align with this setup, we sampled 10 instances in our template. We split these into a deletion set of 5 samples and a learning set of 5 samples. This choice was necessary due to the limited context window of the models used. We then performed in-context unlearning under these constraints.
>
> The results indicate that this method is ineffective with a small context window when dealing with larger deletion sets (128 in our experiment settings). This suggests poor generalization of ICUL in such scenarios.
>
> |125M|EL$_{10}$|MA|BERT|Entropy|
> |:-|:-:|:-:|:-:|:-:|
> |Neo (before unlearning)|51.9|76.8|70.3|4.139|
> |Neo+ICU (ours)|4.4|55.6|53.3|3.833|
> |Neo+ICUL|52.9|73.4|68.5|3.970|
>
> |1.3B|EL$_{10}$|MA|BERT|Entropy|
> |:-|:-:|:-:|:-:|:-:|
> |Neo (before unlearning)|98.2|92.3|86.3|4.640|
> |Neo+ICU (ours)|4.7|51.3|52.7|3.900|
> |Neo+ICUL|91.4|87.4|78.2|4.443|
>
> |2.7B|EL$_{10}$|MA|BERT|Entropy|
> |:-|:-:|:-:|:-:|:-:|
> |Neo (before unlearning)|96.7|93.7|90.2|4.719|
> |Neo+ICU (ours)|4.5|48.3|52.7|3.725|
> |Neo+ICUL|97.0|89.9|81.7|4.576|
>
>
> ## W2. Consideration of larger and more realistic benchmarks
>
> Thank you for your comment and suggestion. However, we argue that using a larger benchmark may not bring much insight to the unlearning task. This is because the training data is much larger than the unlearned data in most cases.
>
> Regarding the benchmark's realism, we believe the one we used is also realistic. In fact, it is part of the real-world training dataset, Pile, and is also used in a real-world unlearning challenge (https://github.com/google-research/lm-extraction-benchmark).
>
> In addition, our method is not tied to any specific dataset or model backbone. It is a flexible framework that can be applied to any model and dataset.
>
>
> [1]. Pawelczyk et al. In-Context Unlearning: Language Models as Few-Shot Unlearners. 2023.
>
> [2]. Chen et al. Unlearn What You Want to Forget: Efficient Unlearning for LLMs. 2023.
>
> [3]. Eldan et al. Who's Harry Potter? Approximate Unlearning in LLMs. 2023.

---

> > ### Comment · Reviewer_7C31 · 2024-12-02
> >
> > Thanks for your response. However, I think this work lacks a comprehensive discussion with previous machine unlearning works and how the proposed method differ from them. Also, I'm still concern about the benchmarks used in this work. Hence, I would like to keep my score.

---

> > > ### Author Response · Authors · 2024-12-02
> > >
> > > Thank you for your time and effort in reviewing our paper.
> > >
> > > In response to your concern about comparing our approach with previous machine unlearning methods, we have addressed the details of these methods in Section 2.1 of the Related Work section.
> > >
> > > Our method differs from earlier approaches, such as KUMPR [1], in a fundamental way. While performing unlearning, our method at the same time allows the model to learn from paired data. Compared to methods that also utilize a retained dataset, such as DPO [2], KL [2], and LLMU [3], our approach takes an iterative strategy. It updates the target data based on the model's current unlearning status. These two strategies enhance the effectiveness of our method in unlearning tasks while preserving the model’s generalization ability. We will add this discussion to the paper for greater clarity.
> > >
> > > As for the methods you mentioned, in-context unlearning [4] is not an ideal comparison. This method does not update model parameters, and as our experiments reveal, it performs poorly with deletion sets with size of 128. EUL [5] introduces additional layers into the transformer architecture. This makes it harder to implement and less adaptable across different model architectures. Eldan's method [6] involves training a reinforcd model on the target data, and this requires the target data to be large enough to train a model, for example the size of the Harry Potter books. At the same time, the authors also mention that applying their technique with the Harry Potter books as the unlearn target may cause the model to forget the wikipedia article and other training data that discusses the books as an unwanted side-effect. These two limitations make the method less practical comapred to our approach.
> > >
> > > Regarding your concern about the benchmark, the one we used is widely used in related works [1,7] and they only use the benchmark as well. We believe that alternative benchmarks, such as WMDP [8], are different datasets designed for the same unlearning task. We have already conducted extensive experiments on the lm-extraction-benchmark. This includes testing three different backbone sizes, and evaluating 9 classification and 4 dialogue downstream tasks (you may refer to Downstream Dataset part in Section 4.1 for details). We have also discussed the generalization of both the backbone and the data in our paper (you may refer to Appendix C for details).
> > > Therefore, we do not understand the specific concerns you have about our choice of benchmark. It is neither practical nor reasonable to ask us to test all possible benchmarks directly.
> > >
> > > [1] Jang et al. Knowledge unlearning for mitigating privacy risks in language models. 2022.
> > >
> > > [2] Maini et al. Tofu: A task of fictitious unlearning for llms. 2024.
> > >
> > > [3] Yao et al. Large language model unlearning. 2023.
> > >
> > > [4] Pawelczyk et al. In-Context Unlearning: Language Models as Few-Shot Unlearners. 2023.
> > >
> > > [5] Chen et al. Unlearn What You Want to Forget: Efficient Unlearning for LLMs. 2023.
> > >
> > > [6] Eldan et al. Who's Harry Potter? Approximate Unlearning in LLMs. 2023.
> > >
> > > [7] Kassem et al. Preserving Privacy Through DeMemorization: An Unlearning Technique For Mitigating Memorization Risks In Language Models. 2023.
> > >
> > > [8] Li et al. The WMDP Benchmark: Measuring and Reducing Malicious Use With Unlearning. 2024.

---

### Official Review · Reviewer_EKcS · 2024-11-07

**Soundness:** 2
**Presentation:** 3
**Contribution:** 2
**Rating:** 5
**Confidence:** 4

**Summary:**

This paper focused on the machine unlearning for decoder-only large language models. Compared with previous unlearning approaches which usually require access to the original training data and harm the model's general capabilities, they propose Iterative Contrastive Unlearning (ICU) to tackle these challenges. Specifically, they first maximize the LM loss for the data to be unlearned. In the meanwhile, they utilize KNN sampling to retrieve similar data and minimize the LM loss and KLD compared to the original model. Lastly, they introduce an iterative process to validate the efficacy of unlearning to provide an appropriate stopping criterion. Through experiments, ICU is showing better unlearning abilities compared with baselines while keeping general LM abilities.

**Strengths:**

1. The retrieved samples and introduced stopping criterion help with the balance between unlearning and keeping original capabilities.

2. They did experiments to validate the effectiveness of ICU, together with studies on parameter sensitivity.

**Weaknesses:**

1. It seems that ICU still require access to a large relevant set (where the relevant data could be retrieved). However, this might not be always available especially when it comes to unlearn privacy-related data. In such cases, it is hard/impossible to retrieve large amount of similar data, which makes ICU less generalizable to more unlearned settings. Recent approach that utilize representation unlearning seems to avoid such data issues ([1]) and the authors seem not discuss with them as well as utilize the benchmark for evaluation.

2. It is unclear how the a and b selected in the ICU to decide the stopping criterion, this part might be tricky and need careful efforts to tune for different types/sets of data, which also make the methods less applicable.

3. It is also unclear the relation ship between loss (3) and (4). Though a set of hyper-parameter experiments are shown, they might want to consider the cases where only (3) or (4) is used. Also, if minimizing KLD is used for learned data, why maximizing KLD is not used of unlearned data? The selection of objective seems to be a bit random.

4. For evaluation, the authors might want to discuss the robustness towards to attacks such as paraphrasing/translation. It is unclear if the model "actually" unlearn the knowledge or just forget the question/instruction/prompt pattern.


[1] The WMDP Benchmark: Measuring and Reducing Malicious Use With Unlearning

**Questions:**

1. How many data is needed for KNN sampling? What is the ratio between unlearned data and retrieved data?

2. Is the model robust to noise from embeddings/KNN sampling?

---

> ### Author Response · Authors · 2024-11-23
>
> ## W1. Concerns about the requirement of relevant datasets for ICU
>
> We appreciate the reviewer's concern regarding the generalizability of ICU in scenarios where access to relevant datasets may be limited. However, we would like to clarify that the RMU method proposed in WMDP [1] also requires access to a dataset distinct from the target forget dataset. Specifically, RMU employs a *forget loss* to align the generated outputs on the forget set with a fixed random variable, and a *retain loss* to ensure that the updated model aligns closely with the frozen model on a benign dataset ($D_{\mathrm{retain}}$). Notably, WMDP utilizes Wikitext as $D_{\mathrm{retain}}$, which serves a similar role to our $D_{\mathrm{lrn}}$ in preserving the generative capabilities of the model.
>
> In our ICU framework, we follow a comparable strategy by leveraging publicly available datasets like Wikitext to construct $D_{\mathrm{lrn}}$. This approach balances unlearning with the preservation of the model’s overall generative performance. Thus, the requirements of ICU in this regard are not more restrictive than those of RMU, as both rely on access to auxiliary datasets for effective model optimization.
>
> ## W2. How a and b are selected
>
> We determined the thresholds for BERTScore ($a = 0.3$) and BLEU ($b = 0.01$) based on preliminary experiments. Specifically, for BERTScore, only 2% of outputs fell below 0.3, while the average score exceeded 0.7, with many instances close to 1. For BLEU, the largest value below 0.01 was 1e-78, and the smallest above 0.01 was 0.0159, making 0.01 a clear dividing point. These thresholds effectively balance precision and applicability based on the observed data distributions.
>
> ## W3. Relationship between loss (3) and (4)
>
> Loss (3) ensures the model generates paired data instead of the original target data, while Loss (4) preserves the generation ability of the original model. For experiments with only (3) or (4), please refer to the ablation study results in Figure 3. Maximizing KLD for unlearned data is unnecessary, as Loss (1) already maximizes the NLL loss for such data. KLD is specifically designed to maintain generative capabilities.
>
> ## W4. Robustness towards attacks such as paraphrasing or translation
>
> Thank you for your valuable suggestion. We conducted experiments to validate this. The results are shown below. When using a paraphrased and translated prefix, the original model achieves comparable metrics. In contrast, the model unlearned using our ICU method shows significantly lower metrics. This demonstrates that the model forgets the unlearned data itself, rather than just the pattern.
>
> |prefix|125M|EL|MA|BERT|
> |:-:|:-|:-:|:-:|:-:|
> |original|Neo (before unlearning)|51.9|76.8|70.3|
> |original|**Neo+ICU (ours)**|4.4|55.6|53.3|
> |paraphrased|Neo (before unlearning)|14.4|66.0|39.1|
> |paraphrased|**Neo+ICU (ours)**|3.7|49.0|37.0|
> |translated|Neo (before unlearning)|28.1|70.0|57.1|
> |translated|**Neo+ICU (ours)**|3.7|51.4|48.9|
>
> |prefix|1.3B|EL|MA|BERT|
> |:-:|:-|:-:|:-:|:-:|
> |original|Neo (before unlearning)|98.2|92.3|86.3|
> |original|**Neo+ICU (ours)**|4.7|51.3|52.7|
> |paraphrased|Neo (before unlearning)|25.2|77.9|45.7|
> |paraphrased|**Neo+ICU (ours)**|1.6|45.1|40.9|
> |translated|Neo (before unlearning)|55.8|83.8|65.2|
> |translated|**Neo+ICU (ours)**|3.6|48.5|51.6|
>
> |prefix|2.7B|EL|MA|BERT|
> |:-:|:-|:-:|:-:|:-:|
> |original|Neo (before unlearning)|96.7|93.7|90.2|
> |original|**Neo+ICU (ours)**|4.5|48.3|52.7|
> |paraphrased|Neo (before unlearning)|27.8|79.7|47.5|
> |paraphrased|**Neo+ICU (ours)**|2.6|41.2|40.7|
> |translated|Neo (before unlearning)|60.9|85.6|67.1|
> |translated|**Neo+ICU (ours)**|4.4|44.9|51.5|
>
>
> ## Q1. How many data is needed for KNN sampling? What is the ratio between unlearned data and retrieved data?
>
> The original benchmark contains 15000 data. In our experiment, we unlearn 128 data at a time. The remaining data in the benchmark is used for sampling. For each instance of unlearned data, we sample one paired data. The ratio between unlearned data and retrieved data is 1:1.
>
> ## Q2. Is the model robust to noise from embeddings/KNN sampling?
>
> We have not conducted specific experiments on this, but it is an interesting question worth exploring. Intuitively, if the retrieved data is random, it would act like using prompts to generate random outputs. While this might affect the unlearned data to some extent, the KL loss can control its overall impact on the model's performance.
>
> [1] Li et al. The WMDP Benchmark: Measuring and Reducing Malicious Use With Unlearning. 2024.

---

> > ### Comment · Reviewer_EKcS · 2024-11-26
> >
> > Thank the authors for the response to the concerns! I will raise the score to 5.

---

> > > ### Author Response · Authors · 2024-11-27
> > > **Thank you**
> > >
> > > Thank you very much for your positive feedback and for raising the score. We sincerely appreciate your insightful comments and recognition of our work.

---

### Author Response · Authors · 2024-11-23

We thank the reviewers for taking the time to review our paper. The detailed comments and thoughtful suggestions have been very helpful. They have significantly improved the quality and clarity of our work. We have revised the manuscript, and to highlight the changes, we have marked them in blue.

The main revisions include:
- Update Equation (5) to differentiate the reference and generated text.
- Modify the notations to ensure consistency throughout the paper.
- Add an additional result in Table 2.
- Provide a more detailed explanation of the Human Evaluation section.

We hope that these changes address the concerns. Please let us know if any further information is needed.

---

### Note · Authors · 2025-02-01

I have read and agree with the venue's withdrawal policy on behalf of myself and my co-authors.